# Fruit Breeding in Regard to Color and Seed Hardness: A Genomic View from Pomegranate

**Xinhui Zhang [1,2], Yujie Zhao [1,2], Yuan Ren [1,2], Yuying Wang [1,2] and Zhaohe Yuan [1,2,*]**

1   Co-Innovation Center for Sustainable Forestry in Southern China, Nanjing Forestry University, Nanjing 210037, China; zhxinhui@njfu.edu.cn (X.Z.); z1184985369@njfu.edu.cn (Y.Z.); renyuan426@njfu.edu.cn (Y.R.); wangyuying@njfu.edu.cn (Y.W.)
2   College of Forestry, Nanjing Forestry University, Nanjing 210037, China
*   Correspondence: zhyuan88@hotmail.com

**Abstract:** Many fruit trees have been whole-genome sequenced, and these genomic resources provide us with valuable resources of genes related to interesting fruit traits (e.g., fruit color, size and taste) and help to facilitate the breeding progress. Pomegranate (Punica granatum L.), one economically important fruit crop, has attracted much attention for its multiple colors, sweet and sour taste, soft seed and nutraceutical properties. In recent years, the phylogenesis of pomegranate has been revised which belongs to Lythraceae. So far, three published pomegranate genomes including 'Taishanhong', 'Tunisia' and 'Dabenzi' have been released on NCBI with open availability. This article analyzed and compared the assembly and annotation of three published pomegranate genomes. We also analyzed the evolution-development of anthocyanin biosynthesis and discussed pomegranate population genetics for soft seed breeding. These provided some references for horticultural crop breeding on the basis of genomic resources, especially pomegranate.

**Keywords:** genomic resources; fruit traits; anthocyanin; soft seed

## 1. Introduction

The growing interests regarding fruit quality and the sudden climate changes require breeders to accelerate the fruit breeding improvements with new efficient breeding approaches, both traditional and unconventional techniques [1–3]. Some popular fruit trees have been whole-genome sequenced, including grape [4], apple [5–7], peach [8,9], strawberry [10,11], pear [12], orange [13], banana [14], pineapple [15], kiwifruit [16,17], blueberry [18], durian [19], waxberry [20], walnut [21,22], mei [23], Chinese chestnut [24], apricot [25], pistachio [26], cherry [27], coconut [28] and pomegranate [29–31]. These genomic resources will help to explore the evolution-development of numerous key horticultural traits (such as the fruit color [32]), and facilitate the breeding progress through developing molecular markers from resequencing projects (e.g., GWAS analyses for apple fruit size [5]; QTL for watermelon taste [33]).

Pomegranate (Punica granatum L.) is an ancient crop with a long domestication history starting since 4000–3000 BC [34] and actually accounts for a total area of more than 6000 km$^2$ globally, especially with increasing cultivation in China, Iran, India, Turkey and America, and with the potential for marketing and mass consumption [29]. Hence, the biology of pomegranate has achieved considerable attention. Firstly, exciting advances in understanding the taxonomy of pomegranate have recently been revised by using whole-genomic analyses [29]. On the basis of morphological [35], molecular [36], genomic [29] evidences, pomegranate is widely accepted as being part of the Lythraceae rather than the Punicaceae family, and its well resolved phylogenetic position provides a key resource for plant phylogenomics. A second debated question is the origin of pomegranate, which was thought to be

originated in Central Asia (Himalaya range) or from Mediterranean area [34,37]. It is adaptable to different climate conditions and has a wide range of global distribution [38]. The most popular attention is regarding its breeding, especially for fruit color, medicinal and pharmaceutical active compounds, and seed softness. Pomegranate possesses more than 500 globally-distributed varieties, and its morphological characteristics, quality and genotype are evaluated by morphological, biochemical and molecular markers providing some evidence for its genetic diversity [39]. It is known as a 'super-fruit' due to its high total phenolic content (e.g., around 1875 and 11,250 mg/L detected in whole pomegranate fruit [40]), especially for the abundant and unique punicalagin (128.02–146.61 mg/g in peel) [41] whose antioxidant activity is 3.11, 9.94 and 39.07 times more than blueberry, sweet orange and apple, respectively [42]. These active compounds have hypolipidemic, antiviral, anticancer, immunomodulation and improving metabolic syndrome function [43–46].

Molecular techniques can reduce the time and costs for genetic profiles screening. Although the screening is in infancy, the adoption of modern sequencing technologies is accelerating cultivar improvement. There are three published pomegranate genomes released on NCBI with open available, including 'Taishanhong' with an assembled size of 274 Mb, 'Tunisia' with an assembled size of 320 Mb and 'Dabenzi' with an assembled size of 328 Mb (Table 1), [29–31]. They also provide us with valuable resources containing numerous genes related to fruit traits of interest with potential roles in developing markers for the molecular-assisted selection (MAS) [29–31].

**Table 1.** Statistics of 'Dabenzi', 'Taishanhong' and 'Tunisia' genome assembly and annotation.

| | Dabenzi | Taishanhong | Tunisia |
|---|---|---|---|
| Sequencing platform | Illumina HiSeq 2000 | Illumina Hiseq 2500 | Pacific Biosciences (PacBio) Sequel platform SMART |
| K-mer | 356.98 | 336.00 | |
| Assembled genome size (Mb) by flow cytometry | 328.13 | 322.70 ± 9.80 | 313.18 |
| Assembly length (Mb) | 328.38 | 274.00 | 320.31 |
| Number of chromosomes (2n) | 18 | 18 | 16 |
| Number of scaffolds | 1111 (≥2 kb) | 2117 (≥1 kb) | 473 |
| Scaffold N50 length (Mb) | 1.89 | 1.70 | 39.96 |
| Longest scaffold (Mb) | - | 7.60 | 55.56 |
| Total size of assembled contigs (Mb) | - | 269.00 | - |
| Number of contigs | - | 7088 (≥1 kb) | 661 |
| Contig N50 length | 66.97 kb | 97.00 kb | 4.49 Mb |
| Longest contig | - | 528.60 kb | 14.77 Mb |
| GC content (%) | 39.40 | 39.20 | 40.38 |
| Percentage of assembly (%) | 94.32 | 94.30 | 97.76 |
| Predicated number of gene models | 29,229 | 30,903 | 33,594 |
| Average gene length (bp) | 2574.61 | 2332.8 | 2229 |
| Average CDS length (bp) | 1077.85 | 1110.40 | 1048.00 |
| Average exon number per gene | 4.31 | 4.52 | - |
| Average exon length (bp) | - | 245.90 | 263.00 |
| Average intron length (bp) | - | 347.60 | - |
| Percentage of contigs anchored on chromosome (%) | 70.32 | - | - |
| Percentage of genes anchored on chromosome (%) | 84.62 | - | 97.76 |
| Pertence of repetitive sequence (%) | 46.10 | 51.20 | 50.93 |
| Percentage of TE to repetitive sequence (%) | 92.62 | 82.10 | 51.80 |
| Percentage of retrotransposons (%) | 40.50 | 35.32 | 24.05 |
| Percentage of DNA transposons (%) | - | 6.35 | 2.33 |
| LTR rate (%) | 17.06 | 17.40 | 24.59 |
| Reference | [30] | [29] | [31] |

## 2. Assembly and Annotation of the Pomegranate Genomes

To date, three published pomegranate genomes 'Dabenzi', 'Taishanhong' and 'Tunisia' have been sequenced on Pacific Biosciences (PacBio) Sequel platform SMART, Illumina Hiseq 2500 and Illumina HiSeq 2000, respectively [29–31]. Among them, 'Dabenzi' and 'Tunisia' assembled to the chromosomal scale, while 'Taishanhong' assembled to the scaffold level. The assembly length of 'Dabenzi' is 328.38 Mb, representing 92.0% of estimated genome size (356.98 Mb) by k-mer, while it is close to the estimated size (328.13 Mb) obtained by flow cytometry. The total size of the assembled sequences of 'Taishanhong' (274 Mb) is 81.5% of its genome size (336 Mb) estimated by k-mer and about 85.0% of its size (322.7 ± 9.8 Mb) is assembled by flow cytometry. The size of the assembled sequences of 'Tunisia' (320.31 Mb) is higher than its size (313.18 Mb) assembled by flow cytometry. The quality of assembled genomes varied with each other, such as 1111 (≥ 2 kb) scaffolds of 'Dabenzi' with an N50 of 1.89 Mb, 2177 scaffolds (≥ 1 kb) of 'Taishanhong' with an N50 of 1.7 Mb, and 473 scaffolds of 'Tunisia' with an N50 of 39.96 Mb. The GC (guanine and cytosine) contents among the three pomegranate genome assemblies are similar, about 39.20–40.38%. They are all similar to the proportion of GC of Eucalyptus grandis (39.29%, [47]). A total of 29,229, 30,903 and 33,594 protein-coding gene models were respectively predicted in 'Dabenzi', 'Taishanhong' and 'Tunisia' genome (Table 1).

In some plant genomes, the major components are repetitive sequences, especially for gymnosperms (53–99% for Pinus [48]), that are mainly caused by the slow and steady accumulation of a diverse set of long terminal repeat (LTR) elements and the paucity of genome rearrangements for adapting to a wide range of ecological niches and evolving multiple defense mechanisms [49,50], contributing to genome stability and evolution [51,52]. The repetitive sequences account for 46.1% (155.3 Mb), 51.2% (140.2 Mb) and 50.93% (163.12 Mb) of 'Dabenzi', 'Taishanhong' and 'Tunisia' assembly genome, and approximately 92.62%, 82.1% and 51.8% of the 'Dabenzi', 'Taishanhong' and 'Tunisia' repetitive sequences were identified as TEs (transposable elements) [29–31]. Retrotransposons and DNA transposons are two major classes of TEs of which the activity contributes to plant evolution and adaption [53]. Among plant genomes, the long terminal repeat elements account for the major proportion of TEs, such as 17.06%, 17.4% and 24.59% for 'Dabenzi', 'Taishanhong' and 'Tunisia', respectively [29–31]. The two main subfamilies of LTR are Copia and Gypsy in pomegranate and its genome was estimated to only experience one more recent expansion of Copia and Gypsy [29]. The expansion of Copia and Gypsy also exists in grape, sunflower and maize [54–56]. Some evidence pointed out that Copia and Gypsy coexisted in the last common ancestor of monocots and dicots [57]. Specific expression of Copia-99 and Gypsy-14 in pomegranate development indicated that the divergent part of LTRs may be related to plant-specific biological processes [29]. The expression of LTR families show positive correlation to gene copy number. Such as in tea, the Ty3/gypsy (~0.94%) expression is two times of that of Ty1/copia (~0.48%) [58]. The nonautonomous large retrotransposon derivations (LARDs) are considered as the remnants of the deletion of autonomous LTR retrotransposons. Ref. [29] revealed that LARD families of pomegranate might have experienced expansion during recent evolution and have the activity of ancient retrotransposon. In addition, pomegranate-lineage-specific gene radiations of LARDs and extensive and unique expression especially in the development of peel and aril indicated LARDs might contribute to fruit coloration and ovule development.

One lineage-specific whole-genome duplication (WGD) event was detected in pomegranate after the paleohexaploidy event that was shared within all eudicots through the distribution of transversions at fourfold degenerate sites of homologous gene pairs within the syntenic blocks [29]. A pan-plant phylogenetic tree bearing 26 rosid species with 28 gene sets suggesting that pomegranate and Eucalyptus grandis have a most recent common ancestor [59]. On the basis of the MCMCtree, the divergence of pomegranate and E. grandis was estimated to occur at approximately 69.6 million years ago [29], after the paleotetraploidy event occurred at 109.9 million years ago in E. grandis [47]. With similar distribution patterns between pomegranate and E. grandis estimated by synonymous substitution rate values of syntenic paralogous genes of the ancient duplications, these findings suggested that the paleotetraploidy event was shared by pomegranate and E. grandis [29]. However, One Thousand Plant Transcriptomes

Initiative [60] put forward that pomegranate shared the paleotetraploidy event with Lagerstroemia indica not with E. grandis. This requires more analysis in the future to verify their relationship.

## 3. The Evolution-Development (Evo-devo) of Anthocyanin Biosynthesis in Pomegranate

Anthocyanins which are known as flavonoids are one of large and widespread groups of plant components. Anthocyanins are important for plant growth and development, as well as being of potential benefit to human health [61–64]. Different species contain diverse kinds of anthocyanins [65,66]. In blueberry juice, 10 anthocyanins are detected and cyanidin-3-*O*-glucoside and delphinidin-3-*O*-glucoside are the major anthocyanins [65]. Only four anthocyanins are detected in cranberry juice and the most abundant anthocyanin is peonidin-3-*O*-glucoside [65]. Pomegranate fruit is a rich source of anthocyanins but different pomegranate cultivars contain anthocyanins which demonstrate great variations and differences [67]. For example, the main anthocyanin of four pomegranates 'Lvbaoshi', 'Hongbaoshi', 'Moshiliu' and 'Taishanhong' is cyanidin 3-glucoside, but this pigment content in 'Moshiliu' is 53.52 mg/100 g, which is a 35-fold and 12-fold increase compared to the concentration of 'Lvbaoshi' and 'Hongbaoshi' [68,69], respectively (Figure 1a). One attraction of pomegranate is the appealing colors that are attributed to various anthocyanins, while the regulation of anthocyanin biosynthesis is a complex process [68–71]. The interaction of internal factors and external environment influence fruit coloration [70,72]. At present, three published pomegranate genomes provide a reference for the comprehensive analysis of the genetic basis of coloring to improve pomegranate breeding [29–31].

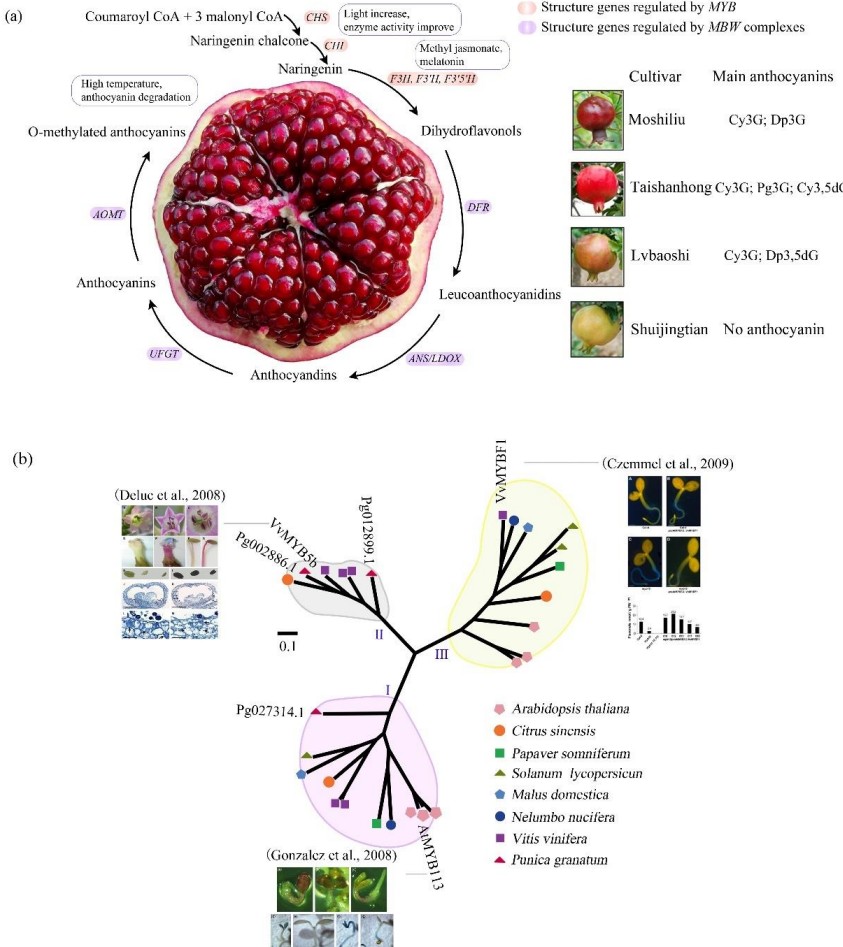

**Figure 1.** (**a**) Model of simplified anthocyanin metabolism pathway of pomegranate. Anthocyanin biosynthetic pathway is a specific branch of flavonoid pathway, which has different regulatory mechanism

in dicot and monocot. EBGs: early biosynthesis genes (including CHS: chalcone synthase; CHI: chalcone isomerase; F3H: flavonoid 3-hydroxylase; F3'H: flavonoid 3'-hydroxylase; F3'5'H: flavanone 3'5'-hydroxylase); LBGs: late biosynthesis genes (including DFR: dihydroflavonol 4-reductase; ANS/LDOX: anthocyanidin synthase/leucoanthocyanidin dioxygenase; UFGT: UDP-glucose: flavonoid glucosyltransferases; AOMT: anthocyanin O-methyltransferase); *MBW* complexes: *MYB-bHLH-WD40* complexes. Methyl jasmonate and melatonin as pre-harvest applications can significantly enhance total anthocyanins of pomegranate with respect to control [73,74]. Dihydroflavonols: Dihydrokaempferol, Dihydroquercetin, Dihydromyricetin; Leucoanthocyanidins: Leucopelargonidin, Leucocyanidin, leucodelphinidin; Anthocyanidins: Pelargonidin, Cyanidin, Delphinidin; Anthocyanins: Pelargonidin-3-glucoside (Pg3G), Cyanidin-3-glucoside (Cy3G), Delphinidin-3-glucoside (Dp3G); O-methylated anthocyanins: Peonidin-3-glucoside, Petunidin-3-glucoside. Cy3,5dG: cyanidin-3, 5-diglucoside; Dp3, 5dG: delphinidin-3, 5-diglucoside. (**b**) Phylogenetic tree constructed with 27 *MYB* genes involved in anthocyanin biosynthesis from eight species. Grape (*VvMYBF1*, *VvMYB5a*, *VvMYB5b*, *VvMYBA1*, *VvMYBA2*); sweet orange (*Cs9g03070.1*, *CsRuby*); apple (*MD16G1158800*, *MdMYB10*); tomato (*Solyc01g079620.2.1*, *Solyc06g009710.2.1*, *SlAN2*); pomegranate (*Pg002886.1*, *Pg012899.1*, *Pg027314.1*); *Arabidopsis thaliana* (*AtMYB11*, *AtMYB12*, *AtMYB111*, *AtMYB75*, *AtMYB90*, *AtMYB113*); *Nelumbo nucifera* (*NnMYB5*); *Papaver somniferum* (*rna-XM_026590419.1*, *rna-XM_026574412.1*).

Three regulatory genes *MYB*, *bHLH* and *WD40* are involved in the anthocyanin biosynthesis pathway [71] (Table 2). Transcription factors that regulate gene encoding determine structural genes' expression intensity, so regulatory genes play crucial roles in plant anthocyanin synthesis. *R2R3-MYBs* are recognized as key transcription factors that regulate anthocyanin accumulation, such as *AtMYB75* (*PAP1*), *AtMYB90* (*PAP2*), *AtMYB113* and *AtMYB114* in *Arabidopsis* [75–77]; *VvMYBA1* and *VvMYBA2* in grape [78,79]; *MdMYB3* and *MdMYB10* in apple [80,81].

**Table 2.** Different regulator genes affecting fruit coloring in some fruit trees.

| Gene | Species | Function | References |
|---|---|---|---|
| *MYB* | Pomegranate | *PgMYB* regulated the accumulation of anthocyanin during reproductive stages. | [82] |
| *MYB1* | Apple | *MdMYB1* regulated genes coordinately in the anthocyanin pathway response to light in apple skin. | [83,84] |
| *MYB3* | Apple | *MdMYB3* regulated the anthocyanin accumulation in apple skin. | [80] |
| *MYB10* | Apple | *MdMYB10* was the key gene that synthesized anthocyanin in red apple fruit. | [81] |
| *MYB110a* | Apple | *MdMYB110a* could up-regulate anthocyanin biosynthesis in apple. | [85] |
| *MYBA* | Apple | *MdMYBA* was a crucial regulator gene in anthocyanin accumulation in red-peel apple induced by low temperature or UV-B irradiation. | [86] |
| *MYB10* | Pear | *PyMYB10* promoted anthocyanin accumulation in fruit fresh and foliage induced by light. | [87] |
| *MYB5a* | Grape | *VvMYB5a* regulated structural genes expression controlling the phenylpropanoid synthesis, such as anthocyanins, flavonols, tannins. | [88] |
| *MYB5b* | Grape | *VvMYB5b* controlled anthocyanin and proanthocyanidin biosynthesis during grape berry development. | [89] |
| *MYBA1* | Grape | *VvMYBA1* could induce red pigmentation when introduced into white-peel grapes. | [79] |
| *MYBA* | Sweet cherry | *PacMYBA* one R2R3-MYB transcription factor from red-colored sweet cherry, played an important role in ABA-regulated anthocyanin biosynthesis. | [90] |
| *MYB10.1* | Sweet cherry | *PavMYB10.1* played a key role in regulating anthocyanin biosynthesis and determined skin color of sweet cherry. | [91] |
| *Ruby* | Blood orange | One MYB transcription factor *CsRuby* contributed to producing anthocyanin induced by cold. | [92] |
| *bHLH3* | Peach | Overexpression of *MYB10.1/bHLH3* and *MYsB10.3/bHLH3* activated anthocyanin production. | [93] |
| *bHLH3* | Apple | *MdbHLH3* bound to *MdMYB1 to* regulate low temperature-induced accumulation. | [94] |
| *bHLH33* | Strawberry | *FvbHLH33*, co-expressed with *FvbHLH33*, strongly activated structural genes in the anthocyanin pathway. | [95] |
| *WD40* | Pomegranate | *PgWD40* with *PgAn1* (bHLH) and *PgAn2* (MYB) co-regulated the downstream structural gene expression involved in the anthocyanin synthesis. | [96] |

Some genomic resources successfully revealed many metabolic pathways and offer molecular markers for agronomics traits [97]. Ono, et al. [98] conducted de novo assembly of the pomegranate

peel transcripts and constructed the anthocyanin metabolic pathway. From pomegranate genome assembly, Yuan, et al. [29] identified 26 anthocyanin biosynthesis candidate genes which showed tissue-specific expression in peel and aril. Moreover, it was found that tandem duplication occurred in *AOMT* (anthocyanin O-methyltransferase) family of pomegranate may evolve new functions like producing anthocyanins. The transcription regulator *MYB-bHLH-WD40* (*MBW*) complexes are the main factors mediating pomegranate peel coloring in the downstream of the anthocyanin pathway [82,96,99]. One R2R3-MYB transcription factor (*PgMYB*) was isolated from pomegranate and characterized that the protein sequence demonstrated high similarity with those of R2R3-family in Arabidopsis, grapevine, eucalytus. *PgMYB* expression reached its peak as the fruit became over-ripened in dark purple-peel pomegranate ('Poost Siyah Yazd') and red-peel pomegranate ('Bozi Isfahan'), while in pink-peel pomegranate ('Shirin Shabad Shiraz'), expression of the gene showed very low levels during the whole development [82]. Gene expression and metabolism analysis of *PgMYB5-like* and *PgbHLH* were performed by agroinfiltrated *Nicotiana benthamiana*. The results showed the two transcription factors interacted to activate flavonoid 3-hydroxylase (F3H), flavonoid 3'-hydroxylase (F3'H) and flavanone 3'5'-hydroxylase (F3'5'H) resulting in the first step of flavonoid production [100]. *PgWD40*, one pomegranate gene is a homologue of *Arabidopsis* TRANSPARENT TESTA GLABRA1 (*TTG1*), could recover the wild-type phenotype with the respect of anthocyanin accumulation of the *Arabidopsis ttg1* mutant. In addition, the total cyanidin content indicated high positive correlation with *PgWD40* expression level. Further expression of *PgWD40*, *An2* (MYB) and *PgAn1* (bHLH) is required to regulate the expression of *PgDFR* and *PgLDOX* in anthocyanin biosynthesis [100]. Zhao, et al. [101] cloned structure genes chalcone isomerase *(CHI)*, chalcone synthase *(CHS)*, *F3H*, dihydroflavonol 4-reductase *(DFR)*, anthocyanidin *(ANS)* and UDP-glucose: flavonoid glucosyltransferases *(UFGT)* from 'Hongbaoshi' with red peel at ripening stage; they did not detect anthocyanin and expression of *ANS* gene in 'Shuijingtian' with white peel at the ripening stage. Compared with *ANS*/leucoanthocyanidin *(LDOX)* in red-peel pomegranate, there was a 90 bp insertion in the first exon region in *ANS/LDOX* of white-peel pomegranate. It was speculated that this insertion sequence was the direct cause of the failure of anthocyanin synthesis in white-peel pomegranate [70]. Luo, et al. [102] identified some different expression proteins and genes which participate in the synthesis of anthocyanins, stilbenoids, diarylheptanoids, gingerols, flavonoids, and phenylpropanoids contributing to the pomegranate peel coloring in two pomegranate cultivars 'Tunisia' bearing mature fruit with red peel and 'White' bearing mature fruit with white peel during fruit coloring at ripening stages.

A maximum-likelihood phylogenetic tree was reconstructed with 27 *MYB* genes involved in anthocyanin synthesis from *Arabidopsis thaliana*, apple, sweet orange, pomegranate, grape, tomato (asterid), *Papaver somniferum* (basal eudicot) and *Nelumbo nucifera* (basal eudicot) (Figure 1b). The R2R3-MYB evolution has been implied to be eudicot lineage-specific [103]. There are three clades, clade I, clade II and clade III, in the phylogenetic tree (Figure 1b). In clade III, some members (such as *ATMYB11*, *ATMYB12*, *ATMYB111* and *VvMYBF1*) have been recognized as regulators for the EBGs (early biosynthesis genes) of anthocyanin synthesis pathway [104,105]. *MYB-bHLH-WD40* (*MBW*) complexes have been recognized to regulate the downstream operators of anthocyanin biosynthesis. The convergent evolution hypothesis for *MBW* complexes implies that co-regulate anthocyanin synthesis [103]. Furthermore, the clade II might be responsible for both EBGs and LBGs (late biosynthesis genes), due to *VvMYB5a*, was confirmed to involve in the regulation of flavonoid and proanthocyanidin accumulation in grape skin, that was responsible for both early and late genes of anthocyanin pathway, except *ANS* [88]; and *VvMYB5b* was recognized to involve in the synthesis of flavonol and proanthocyanidin in grape skin and seed during the flowering stage and the synthesis of anthocyanin in grape berry during harvest [89]. Hence, it was speculated that *Pg012899.1* and *Pg002886.1* in the same cluster might have the similar functions to *VvMYB5a* and *VvMYB5b*. Similarly in the clade I, *AtMYB113* could regulate the production of anthocyanin pigments that was *TTG1-* and *bHLH*-dependent and only regulated the LEBs [76]. *VvMYBA1* and *VvMYBA2* could activate the *VvUFGT* promoter and transport anthocyanin to the vacuole to color the grape skin [79]. *MdMYB10*

induced anthocyanin synthesis with the co-expression genes, *MdbHLH3* and *MdbHLH33* in red fresh apple [81]. In lotus, the over-expression of *NnMYB5* could induce anthocyanin accumulation in flower stalks and upregulate the expression of *TT19* in Arabidopsis [106]. Hence, *Pg027314.1* in this clade might be a regulator for LBGs in the anthocyanin synthesis pathway. In brief, *MYBs* paly roles through the whole anthocyanin pathway and *MYBs* in three clades have diverse functions (Figure 1b). It was speculated that *MYBs* evolved new functions, due to selection of nature for producing different colors to attract birds or other animals to spread seeds or pollen [103].

Besides genetic factors, environmental factors and preharvest treatments could regulate anthocyanin metabolism. Shaked-Sachray, et al. [107] speculated that temperature not only affected the anthocyanin synthesis pathway, but also affected the stability of anthocyanin. The results of grape coloring showed that more concentration of anthocyanin could be accumulated in petals and fruits at lower temperature, while at high temperature, the synthesis rate of anthocyanin slowed down and the degradation rate increased, resulting in a significant decrease in anthocyanin accumulation [108,109]. In addition, temperature could affect genes associated with anthocyanin production at the transcriptional level, such as EBGs including PAL, CHS and CHI and LBGs including DFR and ANS [110,111]. As such, *SmCRYs* inhibited the activity of *SmCop1*, which allowed *SmHY5* and *SmMYB1* to bind to the promoters of *SmCHS* and *SmDFR* genes resulting in the synthesis of anthocyanin in eggplant [112]. Pomegranate trees were preharvest treated with 0.1 mmol/L melatonin along the development growth cycle. The content of total anthocyanins, total phenolics and total antioxidant activity were significantly higher in treated than in non-treated fruits at harvest time and these bioactive compounds remained higher in melatonin-treated fruits after 60 days of postharvest storage [73]. Similarly, preharvest application of methyl jasmonate treatments could improve arils color due to increasing the content of total and individual anthocyanins and enhance fruit crop yield [74].

## 4. Pomegranate Population Genetics for Soft Seed Breeding

Population genetic structure is considered as the amount and distribution of genetic variation within and between populations. The direct manifestation of genetic variation is mutation [113]. To elucidate the evolutionary processes acting on plant populations, we need to discuss levels and distributions of genetic variation, selection and adaptation and genetic structure within and between populations [114].

The origin of pomegranate is disputed. According to Levin [115], three mega-centers (primary, secondary and tertiary) and five macro-centers (Middle Eastern, Mediterranean, Eastern Asian, American and South African) are considered as the origin of pomegranate. Pomegranate has wild and cultivated species. It has been domesticated about 3500 BC in Western Asia [116]. Fluorescent-AFLP (amplified fragment length polymorphism) markers analysis was conducted for 85 pomegranates from six geographical populations located at Shandong, Anhui, Shaanxi, Henan, Yunnan, and Xinjiang Provinces in China. The results indicate that the diversity at a population level is lower than that at a species level, and genetic diversity between populations was significantly different [117]. Molecular variation analysis indicates that most genetic variation in 49 accessions of wild Indian pomegranate is within populations (54%) but not between populations, and gene flow in wild pomegranate accessions is lower than in Chinese pomegranate [117,118]. In Tunisia, the low diversity of pomegranate based on SRR (simple sequence repeats) markers analysis reveals its narrow genetic background due to its limited origins [119]. Sarkhosh, et al. [120] gathered 21 Iran soft-seeded pomegranates and low correlation was detected between fruit characteristics and RAPD (random amplified polymorphic DNA) markers [120].

The divergence between hard- and soft-seeded pomegranates is the embodiment of pomegranate germplasm resource diversity. Hence, it is important to explore the metabolic mechanism of influencing seed hardness in pomegranate. The softness of seeds is a desirable economic trait that enhances the consumptive qualities of fruits, but the complete soft-seeded pomegranate is restricted to a narrow ecological region [121]. In addition, study the formation of the soft seed is beneficial to elucidate the

mechanism of lignin synthesis which contributes to plant growth and development and the build-up of resistance to biotic and abiotic stresses and plant evolution [122,123]. To breed a pomegranate which has the characteristic of soft seed is one of the breeding objectives. However, until then, deciphering the ecological and evolutionary forces that form the population structure of the soft-seeded pomegranate is a priority, and this will help us to understand the emergence of soft seed trait and benefit to crop breeding.

Seed hardness is related to cell wall biosynthesis [124]. By comparing protein expression profiles between two genotype pomegranates ('Tunisia' (soft seed) and 'Sanbai' (hard seed)) at 60 and 120 days after flowering, it was found that 'Tunisia' had lower lignin but higher cellulose biosynthesis [125]. Luo, et al. [126] found that several miRNA-mRNA pairs regulated seed hardness by altering cell wall structure, and mdm-miR164e- and mdm-miR172b-targets included *WRKY*, *MYC* and *NAC1* mainly involving brassinosteroid biosynthesis, cell division and lignin biosynthesis. Luo, et al. [31] summarized that genomic variations and selective genes are two critical factors resulting in the divergence between hard- and soft-seed pomegranates. Twenty-six pomegranate varieties were collected and re-sequenced, and the neighbor-joining tree, population structure, PCA (principal components analysis), and LD (linkage disequilibrium) analyses all indicate support for the clustering of pomegranate clade according to seed hardness. Numerous candidate *MYB*, *WRKY*, *AP2-like*, *MYC* and *NAC* genes with different proportions of SNPs (single nucleotide polymorphisms) and InDels (insertion-deletion) in pomegranate and hawthorns were found to play roles in regulating seed hardness. These transcription factors are involved in brassinosteroid, cell division, lignin, cellulose flavonoid and xyloglucan biosynthesis [31,127,128]. Zarei, et al. [124] found *COMT* exhibited higher expression in soft-seed pomegranate, and *CCR* and *CAD* showed higher expression in hard-seed pomegranate. In the soft-seed variety of hawthorn, four *NAC* and twelve *MYB* TF are all significantly down-regulated [127]. A battery of Secondary Wall-associated NAC Domain Protein (SND1) regulated transcription factors such as *SND2*, *SND3*, *MYB46*, *MYB103*, *MYB85*, *MYB52*, *MYB54*, *MYB42*, *MYB43*, *MYB20*, and *KNAT7* in *Arabidopsis* has been demonstrated to be involved in the process of secondary wall formation and *MYB46* is able to activate the entire secondary wall biosynthesis program [129,130]. Xia, et al. [131] characterized the role of a NAC transcription factor (*PgSND1-like*) involved in the regulation of seed hardness in pomegranate. They found that *PgSND1-like* gene from 'Tunisia' and 'Sanbai' had a single base replacement at the 166-bp position of CDS (coding sequence). Overexpression of *PgSND1-like* ('Sanbai') transgenic *Arabidopsis* could enhance lignin content, while *PgSND1-like* ('Tunisia') in *Arabidopsis* exhibited no phenotypic differences compared with wild type. They speculated that *PgSND1-like* in 'Tunisia' and 'Sanbai' may regulate lignin biosynthesis and seed hardness in pomegranate. The adaptation process of a population following a rapid environmental change or the colonization of a new niche has been reported in two ways—either through new beneficial mutations or through using alleles of the standing genetic variation [132]. The study of phenotypic evolution reveals that strong selection and rapid heritable trait changes in nature are common [133]. All soft-seed pomegranate varieties seem to prefer a tropical Asian climate and hard-seed pomegranates have a stronger ability to endure cold [31]. Examples are that some genes such as *PgL0044640* and *PgL0314990* are enriched in the forkhead box O (FoxO) signaling pathway and *PgL0044700* is involved in the mitogen-activated protein kinase (MAPK) signaling pathway in the hard-seeded population of pomegranate (Table 3) [31]. Besides, in Drosophila melanogaster and tomato, both FoxO and MAPK are involved in cold response [134,135].

**Table 3.** Genes involving the regulation of seed hardness in pomegranate.

| Gene | Function | Reference |
|---|---|---|
| *PgSND1-like* | The overexpression of the *NAC* transcription factor PgSND1-like enhanced lignin concentration in transgenic plants compared with wild-type *Arabidopsis*. | [131] |
| *SUC6* | *SUC6*, one sucrose transport protein, which was more highly expressed at 60 days after flowering than 120 days after flowering in 'Tunisia' and 'Sanbai'. | [126] |
| *SUC8-like* | *SUC8-like* was important for controlling seed development and was down-regulated significantly in soft-seeded pomegranate 'Tunisia' compared to hard-seeded pomegranate 'Sanbai'. | [31,126] |
| *PgL0044640* | These two genes were enriched in the FoxO signaling pathway indicated in the hard-seeded population by KEGG analysis. | [31,134,135] |
| *PgL0314990* | | [31,134,135] |
| *PgL0044700* | This gene was enriched in the MAPK signaling pathway in the hard-seeded population by KEGG analysis. | [31,134,135] |

## 5. Conclusions

Some genomic resources have revealed many metabolic pathways and offer molecular markers for agronomics traits. Many genes in the anthocyanin biosynthesis pathway have been identified, cloned, and their functions partially verified function. We reconstructed a maximum likelihood tree with 27 *MYB* genes related to anthocyanin synthesis. These *MYB* members were divided into the typical three major branches and evolved diverse functions. It was speculated that *MYBs* evolved new functions, due to the natural development of producing different colors to attract birds or other animals to spread seeds. The complexity of regulation of lignin synthesis is the embodiment of soft- and hard-seeded pomegranate germplasm resource diversity. Hence, it is of important significance to study the formation of the soft seed trait. The soft and hard degree of seed in pomegranate is influenced by environmental factors and genetic background. Because of the complexity of metabolic synthesis, additional research will be continued. Besides, many more excellent traits of pomegranate germplasm resources should be explored and exploited to breed pomegranate with various colors and soft seed.

**Author Contributions:** Conceptualization, X.Z. and Z.Y.; Writing—original draft preparation, X.Z.; Writing—review and editing, X.Z., Y.Z., Y.R., Y.W. and Z.Y.; Visualization, X.Z.; Project administration, Z.Y. All authors have read and agreed to the published version of the manuscript.

**Funding:** This research was funded by the Initiative Project for Talents of Nanjing Forestry University [GXL2014070, GXL2018032], the Priority Academic Program Development of Jiangsu High Education Institutions [PAPD], the Natural Science Foundation of Jiangsu Province [BK20180768].

**Conflicts of Interest:** The authors declare no conflicts of interest.

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
