# Peer review of "Fruit Breeding in Regard to Color and Seed Hardness: A Genomic View from Pomegranate"

_agronomy, doi:10.3390/agronomy10070991_

Round 1

Reviewer 1 Report

Review on agronomy-842998

The manuscript involves an overview of the P. granatum genome and details the genetic background of some agronomically important traits. The study is interesting and would be appropriate for the journal but the present version suffers from severe mistakes and deficiencies. There are also a large number of grammar mistakes. I was not able to indicate all of them but the authors must ask for assistance in language editing.

Major considerations:

The description mentions chromosomes of pomegranate but authors forgot to report the chromosome number of the species.

Table 1.

„pertence of repetitive sequence” is not sensible. What does it mean? How is it different to „percentage of TE to total genome size”

It is not explained why repetitive sequences are „especially” major components of gymnosperms’ genomes.

The authors do not give any information on DNA transposons, the MS must be completed with relevant information.

L104: the reasoning regarding the origin of the tetraploid genome is not discussed or assessed properly. The authors should provide some conclusions on different hypotheses.

L132: „ANS/LDOX is a key enzyme which catalyzes the synthesis of colorless anthocyanidins into diverse anthocyanins”: The sentence is incorrect. ANS catalyzes the formation of anthocyanidins that are red or purple in color, anthocyanins are also colorful compounds!

Table 2 lacks important information about sweet cherry MYB and fruit skin color.

L235: „expressed proteins” ???? Genes are expressed, proteins are not.

Some examples of the many minor mistakes that occur abundantly in the MS:

L12: economically important fruit crop

L15: have been released

L16: comparation? There is now such an English word.

L28: describing watermelon as a fruit tree is a bit funny as it is a soft-stem plant

L65 have been…

L89-90: the sentence lacks verb and hence its meaning cannot be found out

L 115: variations instead of variety?

L116: change to „the”

L121: agricultural

L151: Arabidopsis should be set in italics

L184: activate?

Author Response

We have responsed your comments in detail and revised in the paper based on your suggestion.

Reviewer 2 Report

The authors presented very interesting studies about fruit breeding of pomegranate (Punica granatum), an economically important ‘super-fruit’. The multiple color, sweet and sour taste, soft seed and rich nutrition are the most attractive features in term of its breeding. In the study the comparison of the assembly and annotation of three published pomegranate genomes including ‘Taishanhong’, ‘Tunisia’ and ‘Dabenzi’ have been carried out.  The fruit breeding comunity could take advantage from this. Zang and coworkers analyzed the evolution-development of anthocyanin biosynthesis and discussed pomegrant population genetics for soft seed breeding. Generally it is well written manuscript.

A few comments and suggestions are listed below.

  • Ln 134: the references are needed here. Could you please add some informations about this phenomenon in main text.
  • Please increase the informations about use of mutants and/or overexpression lines in pomegranate research if possible.
  • Could you please add the table which summarize the knowledge about the genetic bases of regulation of seed hardness in pomegranate.

Minor comments:

When the first time  the abbreviation of the gene is used it should be explained. Please add it throughout the manuscript.

Ln 11-12: Punica granatum, please use italics for all Latin species name throughout the manuscript

Ln 30: evo-devo please explain the shortcut. It is first time use in the manuscript

Ln 74: please use 29,229; 30,903 instead of 29229 etc. throughout the manuscript and constantly in Table 1

In Table 1 please use  constantly capital letter for Percentage

In Table 1 please add the references about genomes sequencing of these pomegranate varieties

Ln 116: it should be the main not The and four pomegranate not three

Figure 1b: Malus domestica with capital letter

Ln 140: MYB genes should be in italics

Ln 147 and 262: genes names should be in italics

Ln 158: please use Fang and coworkers or Fang et al.

Ln 161: please explain AOMT shortcut

Ln 182 and 186: please explain EBG and LBG shortcuts

Ln 200: lack of space Figure 1b

Ln 214, 221, 223: please explain the AFLP, SSR and RAPD shortcuts

Ln 171 and 241: please use Luo and coworkers or Luo et al.

Ln 244: please explain PCA and LD shortcuts

Ln 248: what means – other biosynthesis?

Ln 251: what exactly genes form MYB and NAC family are involved in process of secondary wall formation?

Ln 255 and 256: lack of space before [118] and [119]

All references have double numbers. All journal names are not in italics, please reformat as per journal requirements.

Author Response

We have responsed to your comments in detail and revised in the paper based on your suggestion.

Reviewer 3 Report

Dear authors,
I have read and revised your manuscript with great interest. I think it is very useful from a scientific point of view, even if in some parts it seems a bit confusing.
I have tried to make some corrections: however, the manuscript needs a thorough revision of the English language.
The contents are very interesting, but the title needs to be revised. In fact the manuscript deals in detail with some pathways, and for this reason the title must be more detailed. You can find my comments on the pdf file that I have attached

Author Response

(The authors gave the same response as above.)

Round 2

Reviewer 1 Report

The MS has clearly benefitted from the revision and is now eligible for the journal.

Reviewer 3 Report

Dear authors, 

I really appreciate your revised form 

Best regards